# MetaScore: A Novel Machine-Learning-Based Approach to Improve Traditional Scoring Functions for Scoring Protein–Protein Docking Conformations

**DOI:** 10.3390/biom13010121

**Published:** 2023-01-06

**Authors:** Yong Jung, Cunliang Geng, Alexandre M. J. J. Bonvin, Li C. Xue, Vasant G. Honavar

**Affiliations:** 1Bioinformatics & Genomics Graduate Program, Pennsylvania State University, University Park, PA 16802, USA; 2Artificial Intelligence Research Laboratory, Pennsylvania State University, University Park, PA 16802, USA; 3Huck Institutes of the Life Sciences, Pennsylvania State University, University Park, PA 16802, USA; 4Bijvoet Centre for Biomolecular Research, Faculty of Science—Chemistry, Utrecht University, Padualaan 8, 3584 CH Utrecht, The Netherlands; 5Center for Molecular and Biomolecular Informatics, Radboudumc, Greet Grooteplein 26-28, 6525 GA Nijmegen, The Netherlands; 6Clinical and Translational Sciences Institute, Pennsylvania State University, University Park, PA 16802, USA; 7College of Information Sciences & Technology, Pennsylvania State University, University Park, PA 16802, USA; 8Institute for Computational and Data Sciences, Pennsylvania State University, University Park, PA 16802, USA; 9Center for Big Data Analytics and Discovery Informatics, Pennsylvania State University, University Park, PA 16823, USA

**Keywords:** protein–protein docking, scoring functions, machine learning, method combination

## Abstract

Protein–protein interactions play a ubiquitous role in biological function. Knowledge of the three-dimensional (3D) structures of the complexes they form is essential for understanding the structural basis of those interactions and how they orchestrate key cellular processes. Computational docking has become an indispensable alternative to the expensive and time-consuming experimental approaches for determining the 3D structures of protein complexes. Despite recent progress, identifying near-native models from a large set of conformations sampled by docking—the so-called scoring problem—still has considerable room for improvement. We present MetaScore, a new machine-learning-based approach to improve the scoring of docked conformations. MetaScore utilizes a random forest (RF) classifier trained to distinguish near-native from non-native conformations using their protein–protein interfacial features. The features include physicochemical properties, energy terms, interaction-propensity-based features, geometric properties, interface topology features, evolutionary conservation, and also scores produced by traditional scoring functions (SFs). MetaScore scores docked conformations by simply averaging the score produced by the RF classifier with that produced by any traditional SF. We demonstrate that (i) MetaScore consistently outperforms each of the nine traditional SFs included in this work in terms of success rate and hit rate evaluated over conformations ranked among the top 10; (ii) an ensemble method, MetaScore-Ensemble, that combines 10 variants of MetaScore obtained by combining the RF score with each of the traditional SFs outperforms each of the MetaScore variants. We conclude that the performance of traditional SFs can be improved upon by using machine learning to judiciously leverage protein–protein interfacial features and by using ensemble methods to combine multiple scoring functions.

## 1. Introduction

Proteins are among the most abundant, structurally diverse, and functionally versatile biological macromolecules. They come in many sizes and shapes and perform a wide range of structural, enzymatic, transport, and signaling functions in cells [1]. However, proteins rarely act alone as their functions are typically mediated by interactions with other molecules, including, in particular, other proteins. Alterations in protein–protein interfaces leading to abnormal interactions with endogenous proteins, proteins from pathogens, or both, are associated with many human diseases [2]. Protein interfaces have therefore become some of the most popular targets for rational drug design [3,4,5]. However, the development of effective therapeutic agents [6,7,8,9] to inhibit aberrant protein interactions requires a detailed understanding of the structural, biophysical, and biochemical characteristics of protein–protein interfaces. The most reliable source of such information comes from X-ray crystallography [10] and nuclear magnetic resonance (NMR), which identify interfaces at the atomic level; alanine scanning mutagenesis, which identifies interfaces at the residue level; mass-spectrometry-based approaches, e.g., chemical cross-linking and hydrogen/deuterium (H/D) exchange, which identify individual interfacial residues [11,12]; NMR-based approaches [13], e.g., chemical shift perturbations, cross-saturation, and H/D exchange, which determine interfaces at the residue or atomic level [14]; and cryo-electron microscopy (cryo-EM), which can directly image large macromolecular complexes in their native hydrated state [15]. However, because of the technical challenges and the high costs and efforts involved, there is still a large gap between the number of known protein–protein interactions and the availability of 3D structures for those [16]. Therefore, there is an urgent need for reliable computational approaches for predicting protein–protein interfaces and complexes.

Against this background, computational docking has emerged as a powerful tool for modeling 3D structures of protein–protein complexes [17]. Given 3D structures or models of putative protein–protein interaction partners, docking aims to generate 3D models of their complex. Docking involves two key steps: the sampling of the interaction space between the protein molecules to generate docked models and the scoring of the docked conformations to distinguish near-native conformations from the sampled conformations. There has been much recent progress on both sampling as well as scoring [18,19].

The scoring functions that have been developed for protein–protein docking can be broadly grouped into several categories [20]: (1) physics-based scoring functions that typically consist of a linear combination of energy terms, e.g., those used in HADDOCK [21], pyDOCK [22], RosettaDock [23], ZRANK [24], IRAD [25], DFIRE [26], DFIRE2 [27], PISA [28], and SWARMDOCK [29]; (2) statistical-potential-based scoring functions such as 3D-Dock [30], DFIRE [26,27], DECK [31], SIPPER [32], and MJ3H [33], which typically convert distance-dependent pair-wise atom–atom or residue–residue contacts’ distributions into potentials; (3) complementarity, e.g., of shape, energy, or physicochemical characteristics [34,35,36,37,38]; (4) interface-connectivity-based scoring functions [39,40]; (5) evolutionary-conservation-based scoring functions, e.g., InterEvScore [41]; and (6) machine-learning-based scoring functions that combine a wide range of features including the residue propensity of interfaces, the contact frequencies of residue pairs, evolutionary conservation, shape complementarity, energy terms, atom pair distance distributions, etc. [42,43,44,45,46,47,48,49,50,51,52]. However, as evident from the results of recent CAPRI competitions [53], there is considerable room for improvement in both sampling and scoring [17,54,55,56].

Against this background, we introduce MetaScore, an approach to scoring docking conformations that combines any existing scoring function with a random forest (RF) [57] classifier trained to discriminate between near-native and non-native structures. The RF classifier utilizes a variety of features of the interface between the proteins in the docked conformation, including interaction-propensity-based, physicochemical, energy-based, geometric, connectivity-based, and evolutionary conservation features. We report the results of experiments on a standard benchmark, the protein–protein docking benchmark Version 5.0 [58] (BM5), which show that MetaScore outperforms the original scoring function when the two are compared using the area under the curve of success rate (ASR) and area under the curve of hit rate (AHR) for the top 10 predicted conformations. We further describe an ensemble method, MetaScore-Ensemble, that combines the score produced by an RF classifier trained using features including scores of several traditional scoring methods and features of interfaces with the averaged score of the original scoring methods. This ensemble approach even outperforms MetaScore using any single original scoring method. We conclude that machine learning methods can complement traditional approaches to scoring docking conformations.

## 2. Materials and Methods

### 2.1. Training Data Set and Preprocessing

We used the protein–protein docking benchmark Version 4.0 (BM4) [59], which has both the bound and unbound structures of protein–protein complexes, for training in our experiments excluding antigen-antibody complexes and non-dimers. The exclusion is because antigen-antibody complexes have biologically different characteristics from other complexes we target in terms of their evolutionary conservation in binding regions. For each of the remaining (cases), decoy models (BM4 decoy set) were generated by HADDOCK running in ab initio mode using center of mass restraints following its standard three-stage docking protocol: rigid body docking, semi-flexible refinement, and water refinement [60]. We then selected cases and their water-refined decoys using the following criteria: (1) A case has at least one decoy with acceptable or better quality (i.e., the interface root mean squared deviation (*i*-RMSD) of the decoy is less than or equal to 4 Å). (Even if there is no acceptable decoy for the case, the bound structure of the case is used for training. However, such a case cannot be used for evaluation of the scoring method.); (2) The number of interface residues in a conformation is greater than or equal to 10. Interfacial residues are determined using an alpha carbon–alpha carbon (CA-CA) distance of 8 Å between two residues belonging to two different proteins in the conformation (a decoy or a bound form). Among the 176 cases in BM4, 63 cases with decoys HADDOCK generated and 45 cases with only bound structures remained. We labeled a decoy *near-native* if its *i*-RMSD relative to the bound form is less than or equal to 4 Å. Otherwise, the decoy was labeled as *non-native*. This process yielded 1221 *near-native* and 35,957 *non-native* conformations. We refer to this set as the BM4 decoy set. However, the proportion of *near-native* and *non-native* conformations is highly unbalanced. There are various techniques to handle the class imbalance in data sets such as under-sampling, over-sampling, and cost-sensitive training [61,62]. We chose a random under-sampling method and under-sampled the *non-native* conformations for each case so that the *near-native* to *non-native* ratio is 1:1 (after testing 1:1, 1:2, 1:4, and 1:8 using 10-fold case-wise cross-validation on the BM4 decoy set, *data not shown*). We chose *non-native* decoys whose *i*-RMSDs are greater than 14 Å for training a model (after searching and testing 4, 8, 14, and 18 Å as cutoffs, *data not shown*). Our final training set consists of 1221 *near-native* models (*i*-RMSD ≤ 4 Å) and 1221 *non-native* models (*i*-RMSD > 14 Å) for 108 cases.

### 2.2. Test Data Set and Preprocessing

For independent testing, we used sets of decoys generated by HADDOCK from the 55 newly added docking cases to the BM5 [58] (BM5 decoy set) and sets of decoys from CAPRI competitions between CAPRI 10 and CAPRI 30 excluding non-dimers (CAPRI score set) [53]. The CAPRI score set consists of decoys generated from different docking programs, which can represent an ideal set for validating scoring functions independently of the docking programs. The decoys and cases from the BM5 decoy set and CAPRI score set were filtered to the same process as that applied to the training data, the BM4 decoy set. The resulting numbers of cases for the BM5 decoy set and CAPRI score set are 9 and 17, respectively. The corresponding numbers for the decoys were 216 *near-native* and 3384 *non-native* conformations and 1115 *near-native* and 3485 *non-native* conformations for the BM5 decoy set and CAPRI score set, respectively.

### 2.3. Comparison with State-of-the-Art Scoring Methods

We used 10 different state-of-the-art scoring functions to test the MetaScore approach: HADDOCK [21], iScore [52], DFIRE [26], DFIRE2 [27], MJ3H [33], PISA [28], pyDOCK [22], SIPPER [32], SWARMDOCK [29], and TOBI’s method (TOBI) [63]. Among them, HADDOCK, DFIRE2, PISA, pyDock, SWARMDOCK, and TOBI are physicochemical-energy-based scoring functions. SIPPER and MJ3H are statistical-potential-based functions. DFIRE is a function based on both physicochemical energy and statistical potential. iScore is a machine-learning-based scoring function using a random walk graph kernel. 

Both iScore and MetaScore rely on machine learning. However, unlike MetaScore, which uses various features of interfaces of *native* and *non-native* protein–protein conformations to train classifiers that discriminate between *native* and *non-native* conformations, iScore utilizes node labeled graphs to incorporate the details of interfaces. Furthermore, MetaScore is an ensemble technique that can be applied to any combination of scoring functions, including iScore.

### 2.4. Evaluation Metrics

The performance of a scoring method to correctly rank decoys based on *i*-RMSD was evaluated using two metrics: The success rate (the percentage of cases that have at least one near-native conformation among the top *N* conformations) and the hit rate (the overall percentage of near-native conformations that are included among the top *N* conformations). Both were calculated for an increasing number of predictions *N* varying between 1 and 400. For easier comparisons, area under the curve of success rate (ASR) and area under the curve of hit rate (AHR) were computed from the plots of corresponding success rate and hit rate, respectively, for *N* between 1 and 400 predictions. We focus on curves of ASRs and AHRs for the top 10 and top 400 predictions because the top 10 decoys are considered for further analysis in the biologists’ perspective [44] and CAPRI [56] competitions also allow them to be submitted for the next evaluation, and because 400 are the total number of decoys HADDOCK generally generates at its final stage for a case. All metrics are normalized between 0 and 1.

### 2.5. MetaScore, a Novel Approach Combining Scores from Machine Learning Classifier-Based Scoring Function with Scores from a Traditional Scoring Function

MetaScore is an approach that combines the random forest (RF)-based score produced from our RF classifier trained using several features with the score from a traditional scoring function.

#### 2.5.1. The RF Classifier

We trained an RF classifier using a diverse set of features of the interfaces between the interacting partners in decoys of our training data set to discriminate between near-native and non-native conformations. Random forest (RF) is an ensemble tree-structured classifier, which is used for a data set with a large number of training data points and input features [57]. A random forest has two hyperparameters, ntrees (the number of trees to grow) and mtry (the number of features randomly selected as candidates at each split in a tree). They were optimized using a grid search approach; the value of ntrees was set from 10 to 500 with a step length of 10 and the value of mtry was set from 1 to 28 with a step length of 3. The hyperparameter optimization accompanies every RF model trained in different situations such as training with different feature sets, combining with different traditional scoring methods, and so on. The trained RF classifier outputs a probability for a decoy being non-native. The lower an RF score for a decoy, the more likely it to be near-native according to the RF classifier.

#### 2.5.2. The Min-Max Normalization within Each Case

Before combining the scores from different scoring functions including the RF score, we normalized the scores of decoys for each case from each scoring function using the Min-Max normalization method. Min-Max normalization scales a list of data from 0 to 1. The minimum value in the data is mapped to 0 and the maximum one in the data is mapped to 1. The strength of this method is that all relationships among the data values can be preserved exactly and that any potential bias is not introduced into the data [64]. However, the Min-Max normalization is vulnerable to outliers in the original data, e.g., scores of decoys, which have clashes. The resulting normalized values may fluctuate with the existence of outliers in the data set. Before applying the Min-Max normalization, we defined values that fall outside two standard deviations of the mean in the data (here, scores of decoys within a case from a scoring method) as outliers. We forced outliers in the upper side of the data to be assigned 1 and those in the lower side to be assigned 0 as a normalized value. Then, we applied the Min-Max normalization into the remaining original data.

A normalized value (*z*) for *x* in a set of decoy scores for a case, *X*, using this method is calculated as follows:$$z=\begin{array}{c} \frac{x-min\left(X\right)}{max\left(X\right)-min\left(X\right)}\; \end{array}$$
where *min*(*X*) and *max*(*X*) are the minimum and maximum values in the *X* given its range excluding outliers.

#### 2.5.3. The Final Score of MetaScore

The final score is obtained by simply averaging the normalized scores of a decoy from the different scoring methods.

### 2.6. Features of MetaScore

We used seven types of features to encode protein–protein interfaces, each of which has been shown to be useful for characterizing properties of protein–protein interface residues [65,66]. We extracted the following features for the binding site formed by the interacting partners in each decoy: (i) raw and normalized scores from each scoring function (score features), (ii) evolutionary features, (iii) interaction-propensity-based features (statistical features), (iv) hydrophobicity (physicochemical feature), (v) energy-based features, (vi) geometric features, and (vii) connectivity features (see below for detail). A decoy is represented by a feature vector formed by its corresponding features.

#### 2.6.1. Raw and Normalized Scores from Each Scoring Function (Score Features)

We included the raw scores and the normalized scores from each scoring function as part of MetaScore features, which are called score features. Because different methods produce scores in different ranges, and even the scores assigned by a single method to decoys from different docking cases are in general incomparable, there is a need to normalize the scores. We applied the Min-Max normalization method to normalize the scores of decoys in each case for each scoring method. Contrary to the normalized score, the original scores from a classical scoring function also contain valuable information such as the size of the interface region [67], the scoring function’s expertise on how to combine its own multiple features related to binding process, and so on. Therefore, it is expected that a combination of original scores and normalized scores can play roles as complementing each other on training a model. We therefore decided to use both original scores and normalized scores.

#### 2.6.2. Evolutionary Features

Binding sites tend to be highly conserved across species [66,68,69]. A scoring function that ranks decoys based on the degree to which their binding sites match the known or predicted binding sites of the target complex produces rankings that tend to place *near-native* conformations above *non-native* ones [52,70]. Therefore, evolutionary conservation scores of interfacial residues in the binding sites are expected to contribute to classifying decoys into *near-native* decoys or *non-native* models.

We used the Position-Specific Scoring Matrix Information Contents (PSSM-ICs) of interfacial residues as conservation scores. PSSM-IC is a measure of the information content for a residue in a PSSM based on Shannon’s uncertainty using prior residue probability and the relative frequency of the residue at a specific protein sequence position [71]. The higher a value of the PSSM-IC of a residue, the more conserved the residue is. The PSSM-ICs are calculated from a result of multiple sequence alignment using PSI-BLAST [72]. We ran PSI-BLAST of BLAST 2.7.1+ against the NCBI nr database (as of 4 February 2018) to retrieve the sequence homologs of each protein sequence using 3 iterations of PSI-BLAST with an e-value cutoff of 0.0001. Based on the length of the protein sequence, we automatically set “query length-specific” parameters, e.g., BLAST substitution matrix, word size, gap open cost, and gap extend cost, according to a guideline provided in the NCBI BLAST user manual (https://www.ncbi.nlm.nih.gov/books/NBK279684/, accessed on 4 February 2018) (see Appendix A). We collected PSSM-ICs only for interfacial residues between the interacting partners for each decoy and aggregated the PSSM-ICs into three types of representative values: the average, minimum, and maximum of the PSSM-ICs for each and both of two proteins in a decoy. In total, 9 features were generated.

#### 2.6.3. Interaction-Propensity-Based Features (Statistical Features)

Previous studies [30,31,73,74,75] have shown that pair-wise amino acid interaction propensities provide useful information about the interaction patterns of amino acids in complexes. We utilized the interaction propensities of amino acid pairs in the interfacial regions of protein–protein complexes, which were precomputed by InterEvScore [41]. The precalculated interaction propensities can be found in a supplementary table in the InterEvScore paper [41]. The interaction propensity of residue *x* and *y*, *IP*(*x*, *y*), was defined as the ratio of the observed frequency in the protein–protein complexes and the expected frequency derived as the random probability to pick the interaction pair of *x* and *y*.

Moreover, we assumed that interaction propensities weighted by conservation scores and/or distances between interfacial residue pairs can be promising features by reflecting evolutionary closeness and geometrical tightness into the interaction propensity. We generated two additional interaction-propensity-based features weighted by only conservation scores (*IP_PSSM_*) and both conservation scores and distances between interfacial residue pairs (*IP_PSSM,Dist_*). For each interfacial residue pair (*x*, *y*) in the *i*th decoy (*D_i_*), which consists of protein A and B, *IP_PSSM_* and *IP_PSSM,Dist_* are defined as:$$IP_{PSSM}\left(x,\;y\right)=\overset{20}{\underset{m=1}{\sum }}\overset{20}{\underset{n=1}{\sum }}IP\left(m,\;n\right)\times PSSM_{A}\left(x,\;m\right)\times PSSM_{B}\left(y,\;n\right)$$
$$IP_{PSSM,Dist}\left(x,\;y\right)=\frac{IP_{PSSM}\left(x,\;y\right)\;}{Dist\left(x,\;y\right)}$$
where *Dist*(*x*, *y*) represents the CA-CA distance between residue *x* in protein A and residue *y* in protein B, *IP*(*x*, *y*) represents the interaction propensity value for a pair of residue *x* and *y* that InterEvScore provides, and *PSSM_A_*(*x*, *m*) is the position-specific score corresponding to the value of the *m*-th amino acid in the 20-element vector for interfacial residue *x* in the PSSM profile from the sequence of protein A. All PSSM values were normalized by the sigmoid function.

Because the sizes of interfaces of different decoys are various, we summarized a list of values for each type of interaction-propensity-based value (*IP*, *IP_PSSM_*, and *IP_PSSM,Dist_*) from interfacial residue pairs in a decoy by summation and averaging, which results in 6 features.

#### 2.6.4. Hydrophobicity (Physicochemical Feature)

Macromolecules’ physicochemical properties play important roles for the forces of attraction or repulsion among them. Among various physicochemical properties, hydrophobicity has been widely used in not only the scoring of docked conformations but also predicting binding sites [76,77,78,79]. Additionally, the role of hydrophobicity in protein folding/unfolding and interactions has been well known [80,81,82]. We assigned the hydrophobicity values of amino acids from the AAIndex [83] database into all the interfacial residues of both proteins in a decoy and averaged them to use as a feature.

#### 2.6.5. Energy-Based Features

Intermolecular energy plays an essential role in molecular binding and its interaction energy mainly consists of van der Waals and electrostatic interactions [84]. We used the van der Waals, electrostatic, and empirical desolvation energies calculated by HADDOCK for a decoy [85]. We adopted both the normalized and raw values of the energy-based features. Using only raw values for training the RF model is unfair because the values assigned to decoys from different docking cases are incomparable. However, using only normalized values can cause the loss of valuable information implied such as the size and the true net energy produced in the interface of each decoy. For each normalized energy feature, we applied the same Min-Max normalization method.

#### 2.6.6. Geometric Features

##### Shortest Distances of Interfacial Residue Pairs

We assumed that a near-native decoy should be a tightly bound form of the proteins and that decoys would have short and uniform distances of interfacial residues between two different proteins if the two proteins form a tight complex. Hence, we used the shortest distances of interfacial residue pairs as features to reflect the principle of shape complementarity for a decoy. Distances between alpha carbon atoms of the two interfacial residue pairs in a decoy were computed and we selected the top 10 shortest distances. The lower the values are, the more compact the decoy. 

##### Convexity-to-Concavity Ratio

The CX value measures the ratio of the volume that atoms occupy within a sphere with a radius of 10 Å to the volume of empty space in the sphere [86]. It has been widely used in previous studies as a protrusion index [65,87]. The smaller a CX value, the more protruding the atom and its 10 Å neighborhood are. We assumed that if the alpha-carbon atoms of interfacial residues in a protein of a decoy protrude, the ones in their partner interfacial residues in another protein of the decoy would be dented in a compact decoy, and vice versa. In this light, higher convexity-to-concavity ratios using CX values for a pair of interfacial residues can indicate that either residue protrudes and the other one is dented. Keeping this in mind, we generated a feature, CX_ratio_(*x*, *y*), modifying the equation to calculate the ratio of CX values of alpha-carbon atoms of each interfacial residue pair (*x*, *y*). 

Let I_A*1*_, I_A*2*_, …, I_A*n*_ denote a set of interfacial residues in a protein A of a decoy. Here, I_A*i*_ where 1 ≤ i ≤ n is an interfacial residue in protein A, where *n* denotes the number of interfacial residues in the protein A. For each interfacial residue pair (I_A*i*_, I_B*j*_) of a decoy, which consists of protein A and B, CX_ratio_(I_A*i*_, I_B*j*_) is defined as:$${\mathrm{CX}}_{\mathrm{ratio}}\left(I_{Ai},I_{Bj}\right)=\frac{max{\left({\mathrm{CX}}_{Ai},{\mathrm{CX}}_{Bj}\right)}^{2}+1}{min{\left({\mathrm{CX}}_{Ai},{\mathrm{CX}}_{Bj}\right)}^{2}+1}$$
where CX_A*i*_ and CX_B*j*_ represent CX values calculated by centering the 10 Å sphere on alpha-carbon atoms of I_A*i*_ and I_B*j*_, respectively.

CX_ratio_(I_A*i*_, I_B*j*_) is larger than or equal to 1. The higher value of CX_ratio_(I_A*i*_, I_B*j*_) can be regarded as evidence that the alpha-carbon atom of I_A*i*_ or I_B*j*_ protrudes and the alpha-carbon atom of the one is dented. The lower values of CX_ratio_(I_A*i*_, I_B*j*_) can be considered that as evidence that both the alpha-carbon atoms of I_A*i*_ and I_B*j*_ protrude or are dented. Those CX-related values obtained are as many as the number of interfacial residue pairs in the decoy. We summarize them as forms of average and standard deviation, which ends up making a couple of features.

##### Buried Surface Area

The buried surface area [85] is one of the HADDOCK-derived features. The buried surface area estimates the size of the interface between two proteins in a protein–protein complex. It can be obtained by calculating the difference between the entire solvent accessible surface area of two unbound proteins and that of a decoy. We used this value as one of the geometric features for training our model. Because the ranges of buried surface area differ by cases, we normalized buried surface area values by apply the Min-Max normalization method described above, excluding outliers.

##### Relative Accessible Surface Area

The relative accessible surface area (rASA) of each interfacial residue was calculated using both its solvent accessible area obtained using STRIDE [88] and the known surface area of the residue [89]. The average of the rASA values of the interfacial residues was used as a feature for a decoy.

##### Secondary Structure

It is well known that particular secondary structures are preferred at protein interfaces [90,91]. To capture the tendency of protein secondary structures to occur in the interface regions, we counted how many times different secondary structures appear in interfacial residues of a decoy structure. We used 7 secondary structure categories: Alpha Helix, 3–10 Helix, PI-Helix, Extended Conformation, Isolated Bridge, Turn, and Coil. Using STRIDE [88], we counted the occurrence of each secondary structure and normalized the occurrence by dividing it by the number of interfacial residues. In total, 7 features of secondary structures for a decoy were generated.

#### 2.6.7. Connectivity Features

To capture the connectivity of interfacial residues and the size of the interface, we added three features: the number of interfacial residue pairs, the total number of interfacial residues, and the link density. The link density feature was implemented as defined in Basu et al. [92], which is a weighted number of interfacial residue pairs by the maximum number of possible links of the interfacial residues between the two different proteins.

## 3. Results

### 3.1. Combination of Scores from the RF Classifier and Scores from HADDOCK Can Improve the Performance of HADDOCK Scoring

To test our hypothesis that combining a machine learning model trained using potent interaction features with an existing scoring function can improve the performance of the original scoring function, we chose HADDOCK firstly as a representative of traditional scoring methods. We compared three scoring methods, HADDOCK, our RF classifier, and our MetaScore approach combining scores from HADDOCK and the RF classifier (MetaScore-HADDOCK) using 10-fold case-wise cross-validation with the training set derived from BM4 [59] (BM4 decoy set) and independent tested procedures with sets of decoys from the newly added cases from BM5 [58] (BM5 decoy set) and the CAPRI score set [53]. In the 10-fold case-wise cross-validation, a set of cases is randomly partitioned into 10 subsets. Of the 10 subsets, all decoys for cases in a single subset are retained as the test data and scored by a scoring method trained with decoys of cases from the remaining subsets. This process is repeated for all single subsets for testing in the cross-validation.

Table 1 and Figure 1 show that MetaScore-HADDOCK has a performance better than or at least comparable to the original method, HADDOCK, for all four performance metrics across all data sets we tested. The RF classifier itself, however, does not outperform HADDOCK for every data set and every evaluation method. Based on the observations, we conclude that the combination of scores from the RF classifier and HADDOCK could improve the scoring performance.

### 3.2. Evaluation of Feature Importance

To train our RF classifier, we used various types of features of protein–protein interfaces that describe the interaction characteristics between a pair of proteins. We evaluated their impact on the performance of the RF classifier using 10-fold case-wise cross-validation and excluding in turn each of the seven feature types (Table 2). 

In the RF classifier, we found that the ASRs for the top 10 and 400 predictions decreased for each feature type removed. Based on the ASR for the top 10 predictions, which is a more focused evaluation metric for scoring methods, all feature types contribute to the performance of the RF classifier. Among the various types, the connectivity features are the features that contribute the best to the RF classifier but evolutionary features are the least contributing. Although the AHRs for the top 10 and 400 predictions are not the best in the RF classifier using all features, the differences of the AHRs across most of the exclusion tests are insignificant in consideration of their standard deviation. We therefore determined to use the RF classifier using all features as our machine-learning-based model.

To see if feature combinations on training a machine learning model also affect MetaScore–HADDOCK’s performance, we evaluated MetaScore-HADDOCK by excluding each type of feature individually as part of the training of the machine-learning-based model. Table 2 shows that the change of feature combinations has relatively little impact on the performance compared to the RF classifier based on the observation that the standard deviations of the four performance measures in the MetaScore-HADDOCK are lower than those in the RF classifier. Based on these results, we conjecture that combining scores from the two different scoring methods, the RF classifier and HADDOCK, helps to reduce the change in the performance subject to changes among subsets of the entire feature set in the RF classifier. Although MetaScore-HADDOCK using all features does not show the best performance, we choose it as a final model because (1) the difference in performance between the best-performing MetaScore-HADDOCK, which is trained without evolutionary features and MetaScore-HADDOCK using all features is not statistically significant within standard deviation and (2) the RF classifier trained with all features has the best performance in terms of ASR, which is the more relevant evaluation metric for scoring functions. This is because we conjecture that the best-performing RF classifier has a higher chance of resulting in a better MetaScore.

### 3.3. Combination of RF Classifier Scores and Scores from Other Scoring Methods Can Improve the Performance of Each Method

To test if the MetaScore approach can be applicable to other methods, not only HADDOCK, we performed the same procedure using nine previously published scoring functions, iScore [52], DFIRE [26], DFIRE2 [27], MJ3H [33], PISA [28], pyDOCK [22], SIPPER [32], SWARMDOCK [29], and TOBI’s method [63], respectively. We obtained scores from the nine methods for decoys in our two data sets, the BM4 decoy set and BM5 decoy set. For each scoring method, we replaced the normalized HADDOCK scores and the raw HADDOCK scores with the normalized scores and raw scores of the respective scoring methods and retrained our model with each set of scores. The resulting combined methods are called MetaScore-iScore, MetaScore-DFIRE, MetaScore-DFIRE2, MetaScore-MJ3H, MetaScore-PISA, MetaScore-pyDOCK, MetaScore-SIPPER, MetaScore-SWARMDOCK, and MetaScore-TOBI, respectively.

The results in Table 3 show that our MetaScore approach for most original scoring methods improves their performance for both the BM4 decoy set, our training set, using 10-fold case-wise cross-validation and the BM5 decoy set, the test set, in terms of ASR and AHR evaluated over the decoys ranked among the top 10 predictions except for AHR of DFIRE using the BM5 decoy set. Moreover, even though the results of three methods (iScore, PISA and MJ3H) using the BM4 decoy set and five methods (HADDOCK, DFIRE, DFIRE2, MJ3H, and PISA) using the BM5 decoy set do not show the improvement in MetaScore in terms of ASR and AHR evaluated for the top 400 decoys ranked, the performances of MetaScore and the original methods are comparable or the decrease in performance is marginal (less than 2.56%) in the independent testing procedure using the BM5 decoy set. (Appendix A).

These results indicate that our proposed method, MetaScore, using a combination of an RF classifier and an existing original scoring method is likely to improve the performance of the original method.

### 3.4. MetaScore-Ensemble Variants Combining MetaScore with Collections of Traditional Scoring Methods Improve on Both

Ensembles of multiple predictive models are known to often outperform individual models [93,94,95]. We constructed MetaScore-Ensemble, which combines MetaScore with several previously published methods: HADDOCK [21], iScore [52], DFIRE [26], DFIRE2 [27], MJ3H [33], PISA [28], pyDOCK [22], SIPPER [32], SWARMDOCK [29], and TOBI’s method [63], which is called the “Expert Committee.” To examine how the performance of MetaScore-Ensemble varies as a function of the performance of members in the ensemble, we used three scoring method groups (“Groups” in Table 4): the higher-performing group (ExpertsHigh), the lower-performing group (ExpertsLow), and the members in the Expert Committee (Experts). ExpertsHigh and ExpertsLow were chosen based on the ASR and AHR for top 10 predictions obtained by 10-fold case-wise cross-validation using the BM4 decoy set, our training set. ExpertsHigh consists of HADDOCK, iScore, DFIRE, MJ3H, and PISA, and the ExpertsLow consists of the others. In addition, we used five ways of aggregating multiple scores (“Approaches” in Table 4), to see the combination effect on MetaScore-Ensemble against each scoring method group:(1)**RF**(**Group**), which is the RF classifier trained using only the raw scores and the normalized scores of members in a group of scoring methods (group);(2)**RF**(**Group + Features**), which is the RF classifier trained using the protein–protein interface features including the raw scores and the normalized scores of members in a group;(3)**Avg**(**Group**), which is a method averaging the normalized scores of members in a group;(4)**Semi-MetaScore-Group**, which is a method combining the score from the RF classifier trained using only the raw scores and the normalized scores of members in a group with the averaged score of the normalized scores of members in the group;(5)**MetaScore-Group**, which is to combine the score from the RF classifier trained using the protein–protein interface features including the raw scores and the normalized scores of members in a group with the averaged score of the normalized scores of members in the group.

We tested fifteen MetaScore-Ensemble methods in total using combinations of three groups and five approaches (Table 4). For example, MetaScore-ExpertsHigh represents one of MetaScore-Ensemble methods, which combines the score of the RF classifier trained using the protein–protein interface features, the raw scores and the normalized scores of members in the ExpertsHigh Group, with the averaged score of the normalized scores of the members in the ExpertsHigh Group.

The comparison results using ASR and AHR on our independent test set, BM5 decoy sets, are shown in Table 5. The curves of success rates and hit rates are shown in Figure 2. We can observe that most of the MetaScore-Ensemble methods perform better than other scoring functions including single traditional methods and MetaScore variants, and that the MetaScore-Experts, which is the MetaScore-Ensemble method using the MetaScore-Group Approach applied to the Experts Group, has the best performance in both ASR and AHR for top 10 predictions.

Moreover, **Avg**(**Group**), applied to three “Groups” (ExpertsHigh, ExpertsLow, and Experts), outperforms each members in each group. Regardless of which “Group” is used, the **Avg**(**Group**) is outperformed by **RF**(**Group + Features**), **Semi-MetaScore-Group**, and **MetaScore-Group**. Moreover, **MetaScore-Group** outperforms not only **Semi-MetaScore-Group** in every “Group” but also each of the MetaScore variants using each member in the corresponding “Group.” In addition, **RF**(**Group + Features**), which incorporates the features of the interfaces for training the RF classifier outperforms **RF**(**Group**), which does not. Taken together, we can conclude that combining methods using any “Approaches” we tested except **RF**(**Group**) outperform individual methods, and that a machine learning model trained with the additional features of interfacial regions outperforms a simple averaging method and a machine learning model not using features for interfacial regions in decoys.

Additionally, regardless of which one of the five “Approaches” is used, “Approaches” using the ExpertsHigh Group outperform ones using the ExpertsLow Group. In Avg(Group), Semi-MetaScore-Group, and MetaScore-Group approaches, the use of the Experts Group outperforms the use of either the ExpertsHigh or ExpertsLow Group. Except for the RF(Group) and Avg(Group), the “Approaches” using all members in the Experts Group were ranked in the top five methods in Table 5. As we expected, we observed that MetaScore-Ensemble methods, which use better-performing members, can outperform ones that use worse-performing members, and that MetaScore-Ensemble methods using more members can perform better than those using less members, except for MetaScore-Ensemble methods using only an RF classifier.

## 4. Discussion

MetaScore offers a new approach to scoring docking models. MetaScore combines an RF classifier trained to distinguish near-native conformations from non-native decoys using features of their interfaces with a traditional scoring method (by simply averaging their respective scores). We have shown that MetaScore improves upon on each of the published scoring functions including HADDOCK, iScore, DFIRE, DFIRE2, MJ3H, PISA, pyDOCK, SIPPER, SWARMDOCK, and TOBI’s method. This suggests that the interface features carry information beyond that captured by the traditional scoring functions for distinguishing native conformations from non-native decoys.

Not surprisingly, the results of our experiments show that ensembles of better-performing scoring functions outperform individual scoring functions. The superior performance of the ensembles that combine more better-performing scoring functions over those that combine fewer scoring functions suggests that the different individual scoring functions carry complementary information for distinguishing native conformations from non-native decoys.

The results of our experiment point to several possible directions for further improving the performance of MetaScore: (1) using better-performing machine learning methods and/or utilizing more informative features to improve the performance of classifiers trained to distinguish between near-native conformations from non-native decoys; (2) increasing the quantity and quality of the data used to train the classifiers; (3) the use of more effective ensemble methods [93,95] for combining the scores produced by individual scoring methods.

## 5. Conclusions

We have proposed a new approach, MetaScore, to score docking models. MetaScore combines an RF classifier trained to distinguish near-native conformations from non-native decoys using features of their interfaces with a traditional scoring function. The results of our experiments on standard docking benchmarks show that, in the case of each of the scoring functions we tested from the literature, the combination of the RF classifier and the individual scoring function consistently outperforms the individual scoring function. Furthermore, MetaScore-Ensemble, which combines MetaScore with a set of traditional scoring methods, improves upon not only each of original scoring methods but also MetaScore in combination with each individual scoring method in terms of success rate and hit rate (evaluated over the conformations ranked among the top 10). We conclude that ensemble methods that combine multiple scoring functions offer promising ways to leverage the complementary strengths of the individual scoring functions.

## Figures and Tables

**Figure 1 biomolecules-13-00121-f001:**
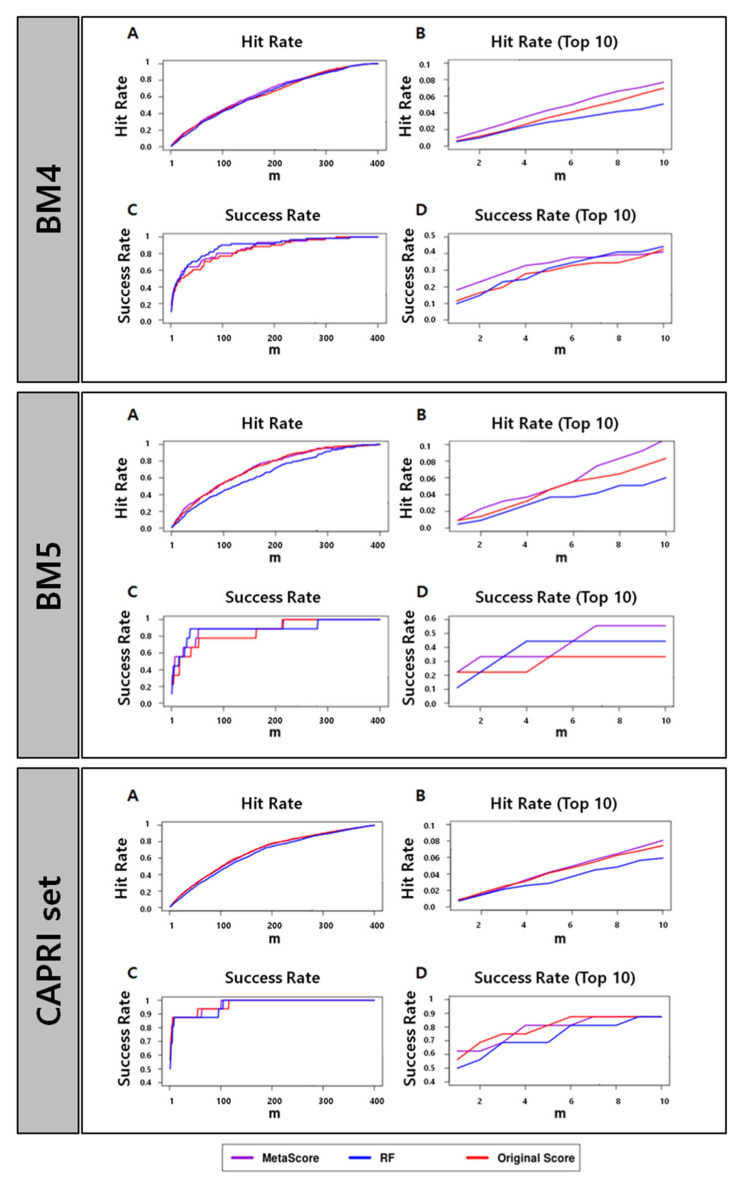
Success rates and hit rates plotted against the top m conformations for a classical scoring method (HADDOCK), machine-learning-based method using RF (RF), and the combined method of the two methods (MetaScore) using the BM4 decoy training set, BM5 decoy set, and CAPRI set (top, middle, and bottom panel, respectively). There are four subpanels for each panel: (**A**) hit rates for conformations of top m ranging from 1 to 400; (**B**) hit rates for conformations of top m ranging from 1 to 10; (**C**) success rates for conformations of top m ranging from 1 to 400; (**D**) success rates for conformations of top m ranging from 1 to 10.

**Figure 2 biomolecules-13-00121-f002:**
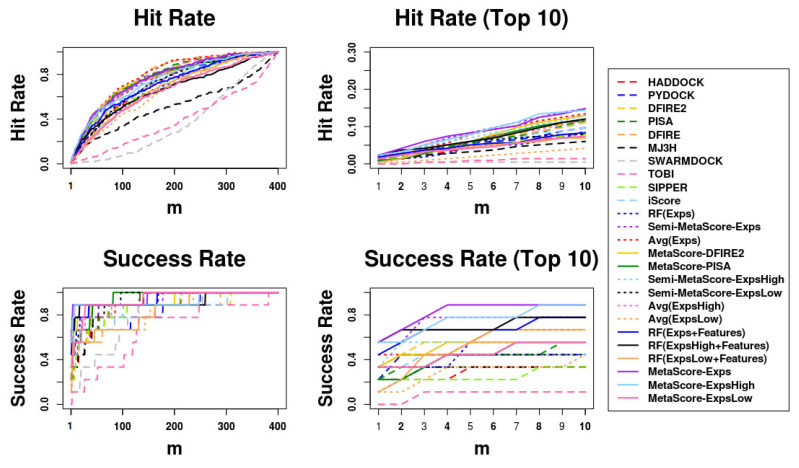
Success rates and hit rates plotted against the top m conformations for original methods, machine-learning-based scoring methods combined with each original method, the averaging method of the Expert Committee’s scores, and machine-learning-based scoring method using the Expert Committee’s scores combined with the averaging method of their scores using BM5 decoy set. The Expert Committee has three groups, the high-ranked group (ExpsHigh), the low-ranked group (ExpsLow), and the group of entire members (Exps). There are four panels: (Top left) hit rates for conformations of top m ranging from 1 to 400; (Top right) hit rates for conformations of top m ranging from 1 to 10; (Bottom left) Success rates for conformations of top m ranging from 1 to 400; (Bottom right) success rates for conformations of top m ranging from 1 to 10.

**Table 1 biomolecules-13-00121-t001:** Performance comparison of three methods, a classical scoring method (HADDOCK), machine-learning-based scoring method using RF (RF classifier), and the combined method of the two methods (MetaScore-HADDOCK) using the BM4 decoy training set, BM5 decoy set, which is a set of decoys generated by HADDOCK from the newly added docking cases to the protein–protein docking benchmark Version 5.0, and CAPRI score set [53].

Data Sets	Method	ASR for Top 10	AHR for Top 10	ASR for Top 400	AHR for Top 400
BM4 decoy set	HADDOCK	0.29	0.036	0.85	0.64
	RF classifier	0.36	0.040	0.89	0.65
	MetaScore-HADDOCK	0.33	0.044	0.87	0.66
BM5 decoy set	HADDOCK	0.29	0.048	0.86	0.72
	RF classifier	0.38	0.032	0.89	0.65
	MetaScore-HADDOCK	0.44	0.056	0.9	0.72
CAPRI score set	HADDOCK	0.80	0.044	0.97	0.68
	RF classifier	0.72	0.032	0.97	0.65
	MetaScore-HADDOCK	0.80	0.044	0.97	0.68

**Table 2 biomolecules-13-00121-t002:** Scoring results by subtracting each feature type.

Method	Excluded Feature Type	ASR for Top 10 ^1^	AHR for Top 10	ASR for Top 400	AHR for Top 400
RF classifier	Connectivity features	0.28	0.044	0.87	0.65
	Statistical features	0.29	0.044	0.87	0.66
	Geometric features	0.30	0.044	0.87	0.66
	Score features	0.31	0.044	0.85	0.65
	Energy-based features	0.32	0.036	0.88	0.62
	Physicochemical feature	0.32	0.044	0.88	0.64
	Evolutionary features	0.34	0.048	0.87	0.65
	None	0.36	0.040	0.89	0.65
MetaScore– HADDOCK	Connectivity features	0.32	0.048	0.86	0.68
	Statistical features	0.34	0.044	0.86	0.67
	Geometric features	0.31	0.048	0.85	0.67
	Score features	0.32	0.048	0.86	0.67
	Energy-based features	0.34	0.048	0.86	0.66
	Physicochemical feature	0.31	0.048	0.85	0.65
	Evolutionary features	0.34	0.052	0.86	0.68
	None	0.33	0.044	0.87	0.66

^1^ The results are ordered by decreased amount of ASR for top 10 in the RF classifier for each exclusion of feature types.

**Table 3 biomolecules-13-00121-t003:** Performance comparison of before and after combining classical scoring methods with each of their corresponding RF classifiers using the BM4 decoy training set and BM5 decoy set, which is a set of decoys generated by HADDOCK from the newly added docking cases to the protein–protein docking benchmark Version 5.0. Our MetaScore approach improved the performance of all scoring functions we evaluated. Numbers in parentheses indicate percentages of increase from original methods. Values with no increase are highlighted in bold.

Data Sets	Method	MetaScore Method				Original Method			
		ASR Top 10	AHR Top 10	ASR Top 400	AHR Top 400	ASR Top 10	AHR Top 10	ASR Top 400	AHR Top 400
BM4	HADDOCK	0.33 (15.28%)	0.044 (22.22%)	0.87 (2.35%)	0.66 (3.13%)	0.29	0.036	0.85	0.64
	iScore	0.48 (4.34%)	**0.074** (**−10.84%**)	**0.89** (**0.00%**)	**0.71** (**−2.74%**)	0.46	0.083	0.89	0.73
	DFIRE	0.32 (29.03%)	0.052 (44.44%)	0.84 (6.33%)	0.67 (3.08%)	0.25	0.036	0.79	0.65
	DFIRE2	0.27 (51.11%)	0.044 (57.14%)	0.85 (4.94%)	0.67 (6.35%)	0.18	0.028	0.81	0.63
	MJ3H	0.38 (9.20%)	0.056 (7.69%)	0.87 (2.35%)	**0.68** (**−1.45%**)	0.35	0.052	0.85	0.69
	PISA	0.42 (6.12%)	0.064 (14.29%)	**0.89** (**0.00%**)	0.71 (1.43%)	0.39	0.056	0.89	0.70
	pyDOCK	0.23 (42.50%)	0.028 (75.00%)	0.81 (8.00%)	0.63 (8.62%)	0.16	0.016	0.75	0.58
	SIPPER	0.26 (128.57%)	0.024 (100.00%)	0.88 (7.32%)	0.62 (10.71%)	0.11	0.012	0.82	0.56
	SWARMDOCK	0.27 (103.03%)	0.028 (133.33%)	0.86 (3.61%)	0.61 (8.93%)	0.13	0.012	0.83	0.56
	TOBI	0.14 (133.33%)	0.012 (200.00%)	0.84 (12.00%)	0.54 (22.73%)	0.06	0.004	0.75	0.44
BM5	HADDOCK	0.44 (52.79%)	0.056 (16.67%)	0.90 (4.65%)	**0.72** (**0.00%**)	0.29	0.048	0.86	0.72
	iScore	0.42 (27.27%)	0.059 (47.5%)	0.86 (13.16%)	0.72 (1.41%)	0.33	0.040	0.76	0.71
	DFIRE	0.49 (2.52%)	**0.064** (**0.00%**)	0.92 (6.98%)	**0.77** (**−1.28%**)	0.48	0.064	0.86	0.78
	DFIRE2	0.57 (8.40%)	0.072 (12.50%)	0.93 (6.90%)	**0.76** (**0.00%**)	0.52	0.064	0.87	0.76
	MJ3H	0.53 (70.51%)	0.056 (55.56%)	**0.91** (**0.00%**)	0.57 (9.62%)	0.31	0.036	0.91	0.52
	PISA	0.43 (1.89%)	0.064 (6.67%)	0.95 (3.26%)	**0.76** (**−2.56%**)	0.42	0.060	0.92	0.78
	pyDOCK	0.56 (31.13%)	0.068 (21.43%)	0.92 (10.84%)	0.77 (2.67%)	0.42	0.056	0.83	0.75
	SIPPER	0.43 (68.75%)	0.060 (25.00%)	0.90 (4.65%)	0.72 (4.35%)	0.26	0.048	0.86	0.69
	SWARMDOCK	0.22 (154.55%)	0.016 (300.00%)	0.86 (8.86%)	0.50 (35.14%)	0.09	0.004	0.79	0.37
	TOBI	0.23 (163.64%)	0.028 (250.00%)	0.78 (16.42%)	0.47 (20.51%)	0.09	0.008	0.67	0.39

**Table 4 biomolecules-13-00121-t004:** Category of scoring method groups and combination approaches for testing MetaScore-Ensemble methods.

Groups of Scoring Functions (Group)	Combination Approaches (Approach)
ExpertsHigh	RF(Group) ^2^
	RF(Group + Features ^7^) ^3^
ExpertsLow	
	Avg(Group) ^4^
	Semi-MetaScore-Group ^5^
Experts ^1^	
	MetaScore-Group ^6^

**^1^ Experts:** Ten published methods including HADDOCK, iScore, DFIRE, DFIRE2, MJ3H, PISA, pyDOCK, SIPPER, SWARMDOCK, and TOBI’s method; **ExpertsHigh**: HADDOCK, iScore, DFIRE, MJ3H, and PISA; **ExpertsLow**: DFIRE2, pyDOCK, SIPPER, SWARMDOCK, and TOBI’s method. **^2^ RF**(**Group**)**:** the RF classifier trained using only the raw scores and the normalized scores of members in a group; **^3^ RF**(**Group + Features**)**:** the RF classifier trained using our feature set of the protein–protein interfaces including the raw scores and the normalized scores of members in a group; **^4^ Avg**(**Group**)**:** a method averaging the normalized scores of members in a group; **^5^ Semi-MetaScore-Group:** a method combining the score from the RF classifier trained using only the raw scores and the normalized scores of members in a group with the averaged score of the normalized scores of members in the group; **^6^ MetaScore-Group:** a method combining the score from the RF classifier trained using our feature set of the protein–protein interfaces including the raw scores and the normalized scores of members in a group with the averaged score of the normalized scores of members in the group. **^7^ Features:** interaction features including score features, evolutionary features, statistical features, physicochemical features, energy-based features, geometric features, and connectivity features.

**Table 5 biomolecules-13-00121-t005:** Performance comparison of scoring methods including original methods, RF classifier variants, averaging method variants, MetaScore variants, and Semi-MetaScore variants using BM5 decoy set, which is a set of decoys generated by HADDOCK from the newly added docking cases to the protein–protein docking benchmark Version 5.0.

Methods	ASR for Top 10 ^1^	AHR for Top 10	ASR for Top 400	AHR for Top 400
MetaScore-Experts	0.82	0.088	0.96	0.77
MetaScore-ExpertsHigh	0.76	0.088	0.93	0.75
Semi-MetaScore-Experts	0.73	0.088	0.94	0.76
RF(ExpertsHigh + Features)	0.70	0.068	0.92	0.66
RF(Experts + Features)	0.67	0.052	0.94	0.71
MetaScore-DFIRE2	0.57	0.072	0.93	0.76
Avg(Experts)	0.57	0.080	0.93	0.80
MetaScore-pyDOCK	0.56	0.068	0.92	0.77
RF(ExpertsHigh)	0.54	0.060	0.89	0.72
MetaScore-MJ3H	0.53	0.056	0.91	0.57
Semi-MetaScore-ExpertsHigh	0.53	0.076	0.90	0.69
Avg(ExpertsHigh)	0.53	0.064	0.90	0.78
DFIRE2	0.52	0.064	0.87	0.76
MetaScore-DFIRE	0.49	0.064	0.92	0.77
DFIRE	0.48	0.064	0.86	0.78
MetaScore-ExpertsLow	0.46	0.044	0.94	0.65
RF(ExpertsLow + Features)	0.46	0.044	0.84	0.68
RF(Experts)	0.45	0.044	0.93	0.67
MetaScore-HADDOCK	0.44	0.056	0.90	0.72
MetaScore-PISA	0.43	0.064	0.95	0.76
MetaScore-SIPPER	0.43	0.060	0.90	0.72
pyDOCK	0.42	0.056	0.83	0.75
PISA	0.42	0.060	0.92	0.78
MetaScore-iScore	0.42	0.059	0.86	0.72
Semi-MetaScore-ExpertsLow	0.41	0.056	0.94	0.72
RF(Features)	0.38	0.032	0.89	0.65
iScore	0.33	0.040	0.76	0.71
MJ3H	0.31	0.036	0.91	0.52
RF(ExpertsLow)	0.31	0.024	0.85	0.64
HADDOCK	0.29	0.048	0.86	0.72
Avg(ExpertsLow)	0.29	0.020	0.84	0.65
SIPPER	0.26	0.048	0.86	0.69
MetaScore-TOBI	0.23	0.028	0.78	0.47
MetaScore-SWARMDOCK	0.22	0.016	0.86	0.50
TOBI	0.09	0.008	0.67	0.39
SWARMDOCK	0.09	0.004	0.79	0.37

^1^ The results are ordered by ASR for top 10 predictions. Note: Features, Group, RF(Group), RF(Group + Features), Avg(Group), Semi-MetaScore-Group, and MetaScore-Group are defined in Table 4.

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
