# Peer review of "MetaScore: A Novel Machine-Learning-Based Approach to Improve Traditional Scoring Functions for Scoring Protein–Protein Docking Conformations"

_biomolecules, 2023, doi:10.3390/biom13010121_

Round 1

Reviewer 1 Report

Query1: when the authors described about the "Energy-based features" as mentioned in 2.6.5 line no 295, they have described a few. to my knowledge in protein docking studies there exists several parameters that contribute to energies. they are binding energy, electrostatic energy, weal vand er waals energy, energy due to unfavourable contributions, intermolecular energy, energy due to inhibitory constant, ligand efficiency.  

my question is that does all these parameters are inculded in your set of docking procedures and are accounted?

Query2: the hydrophobicity parameter involves several interactions? which of those are predominant and what is the combined effect of this on native and dentaured proteins. definitely there will be some variations. how the authors would account for?

Query3: an amino acid does have varying hydrophobicity in proteins since the location as buried or exposed form results difference in the binding inetractions. how could this be solved.

Query4:   2.6.6.3. Buried surface area. The buried surface area (BSA)- this is a wrong terminology in protein studies. THROUGHOUT the field of protein studies BSA refers to bovine serum albumin. provide alternate abbreviations.

kindly also provide abbreviations for the various docking techniques in supplementary section . (introduction section and elsewhere in the article )

Query 5: TABLE 3: the table format is invisible for certain parameters. kindly provide an alternate one 

 Query 5: 3.4. Many heads are better than one- although the titel seems different, i suggest the authors to replace it with better form so that it could be more meaningfull in scientific context

general comments: 

the authors could provide the limitations of this docking methods also. 

Reviewer 2 Report

Protein-protein interactions (PPI) are important in terms of biological functions. The interface between proteins is the main target for main drugs. Thus, docking for PPI is vital for protein-protein complex modelling. In this work, the authors present MetaScore, a random forest-based machine learning model to improve docking scores. This model is trained with multiple physical and chemical features, along with the scores of other scoring functions. The results show that MetaScore achieved a better performance in success rate and hit rate. The work is worth publication with the following points being addressed properly.

(1) For the training set, the authors used undersampling to balance the number of positive and negative samples.  However, the way the author conducted the undersampling is not clear. Is it random undersampling or something else? Also, the authors should discussion the possibility/feasibility of tuning class weights. For example, in sklearn random forest classifier, one can adjust the ‘class_weight’, and in XGBoost model, one can tune ‘scale_pos_weight’ to use the full dataset. See reference below as an example: Hao Tian et al 2021 Mach. Learn.: Sci. Technol. 2 035015.

(2) Since figure 1, 2 and 3 display the same metrics in different datasets, maybe the authors can consider combining them into one large figure. 

(3) Table 3 is not displayed well.

(4) (optional) Maybe the authors can consider releasing their codes publicly on GitHub or other platforms to facilitate related research.

Round 2

Reviewer 1 Report

the authors have justified for the queries raised 

the manuscript can be accepted however 

there are minor modifications needed in the reference section.

the abbreviations of the journals in ref no 20, 71, 80, 82, 90 and 94

Reviewer 2 Report

The authors have addressed all my concerns properly. The work is now worth publication.